# PROCEEDINGS A

materials science, mathematical modelling, statistics

3D printing, Bayesian uncertainty quantification, elastoplasticity, probabilistic mechanics, stochastic finite elements

**Author for correspondence:**
T. J. Dodwell
e-mail: tdodwell@turing.ac.uk

# A data-centric approach to generative modelling for 3D-printed steel

T. J. Dodwell[1,2], L. R. Fleming[3], C. Buchanan[2,4], P. Kyvelou[4], G. Detommaso[5], P. D. Gosling[6], R. Scheichl[7], W. S. Kendall[8], L. Gardner[2,4], M. A. Girolami[2,9] and C. J. Oates[2,3]

[1]Institute of Data Science and AI, University of Exeter, Exeter EX4 4QJ, UK
[2]The Alan Turing Institute, London NW1 2DB, UK
[3]School of Mathematics, Statistics and Physics, Newcastle University, Newcastle upon Tyne, NE1 7RU, UK
[4]Department of Civil and Environmental Engineering, Imperial College London, London SW7 2AZ, UK
[5]Amazon Core AI, Berlin, Germany
[6]School of Engineering, and [7]Institute of Applied Mathematics, Heidelberg University, Heidelberg 69120, Germany
[8]Department of Statistics, University of Warwick, Coventry CV4 7AL, UK
[9]Civil Engineering, University of Cambridge, Cambridge CB2 1PZ, UK

TJD, 0000-0003-0408-200X; WSK, 0000-0001-9799-3480; LG, 0000-0003-0126-6807

The emergence of additive manufacture (AM) for metallic material enables components of near arbitrary complexity to be produced. This has potential to disrupt traditional engineering approaches. However, metallic AM components exhibit greater levels of variation in their geometric and mechanical properties compared to standard components, which is not yet well understood. This uncertainty poses a fundamental barrier to potential users of the material, since extensive post-manufacture testing is currently required to ensure safety standards are met. Taking an interdisciplinary approach that combines probabilistic mechanics and uncertainty quantification, we demonstrate that intrinsic variation in AM steel can be well described by a generative

statistical model that enables the quality of a design to be predicted before manufacture. Specifically, the geometric variation in the material can be described by an anisotropic spatial random field with oscillatory covariance structure, and the mechanical behaviour by a stochastic anisotropic elasto-plastic material model. The fitted generative model is validated on a held-out experimental dataset and our results underscore the need to combine both statistical and physics-based modelling in the characterization of new AM steel products.

## 1. Introduction

Additive manufacturing (AM) technologies developed for metallic material present a rich vein of opportunities in engineering [1], for example building fuel nozzles within jet engines [2], air conditioning unit components and brackets within passenger compartments of airliners [3]. The automotive, medical and aerospace industries are anticipated to account for 84% of the AM market by 2025 [4]. Of the emerging technologies for metallic AM, a prime candidate for producing large-scale components is wire and arc additive manufacturing (WAAM), which is part of the wider field of directed energy deposition (DED). This approach uses robotic arms and arc welding tools, as shown in figure 1a, in such a way that the final object is formed entirely from deposited weld material. WAAM allows for relatively high deposition rates (approx. $4 \, \text{kg h}^{-1}$), good structural integrity, and can be used to produce virtually unlimited part sizes [7], as evidenced by the pedestrian bridge built by Dutch startup MX3D (figure 1b). WAAM is cheaper than other metal AM processes, due to the use of standard off-the-shelf equipment, mature technology, and low-cost wire feedstock [8], making it well-suited to cost-sensitive applications, such as in the construction industry.

The finite precision of WAAM, combined with the complex thermal deformations that occur during welding [9], means that the as-manufactured material properties of WAAM can be considered to be uncertain/stochastic. This is apparent both in the geometry of the material, which is heterogeneous with a banding structure clearly visible (figure 1c), and in the anisotropic physical properties of the material, as shown in this paper (figure 6). The absence of an accurate material characterization for WAAM precludes the use of standard techniques to search for an optimal, resource-efficient design [10], since the performance of a component is intrinsically stochastic and has yet to be accurately quantified. Moreover, the intrinsic variation in components produced by WAAM currently requires that extensive post-manufacture testing is performed, which limits the potential of this material. For example, the aforementioned pedestrian bridge was subject to months of extensive testing by a subset of the authors [5].

The present paper removes the current barriers to manufacturing with WAAM, by reporting the first *statistical* characterization of this novel material. Specifically, we develop a generative statistical model that enables ensemble-based predictions of the performance of a stainless steel WAAM component before it is manufactured.

Our inter-disciplinary approach is a synthesis of *probabilistic mechanics* [11], introducing random variables into traditional finite element descriptions of mechanical behaviour, and *uncertainty quantification* [12], where formal statistical inference techniques are used to derive suitable distributions for the random variables involved. Probabilistic mechanical models are usually posited or elicited [13,14], as opposed to being formally *trained* on a dataset; a key innovation of our work is to use state-of-the-art methods from uncertainty quantification to train a probabilistic mechanical model. Such a data-centric approach is essential, since changes to a printing protocol have the potential to change the nature of variation in material properties, requiring a new statistical characterization to be produced. The power to train probabilistic mechanical models from data is therefore an essential prerequisite to unlocking the full potential of WAAM, removing the main barrier between design and manufacturing with this material.

Previous attempts to characterize the performance of AM material have typically aimed at explaining bulk properties (mechanical strength, fatigue strength, etc). These studies have

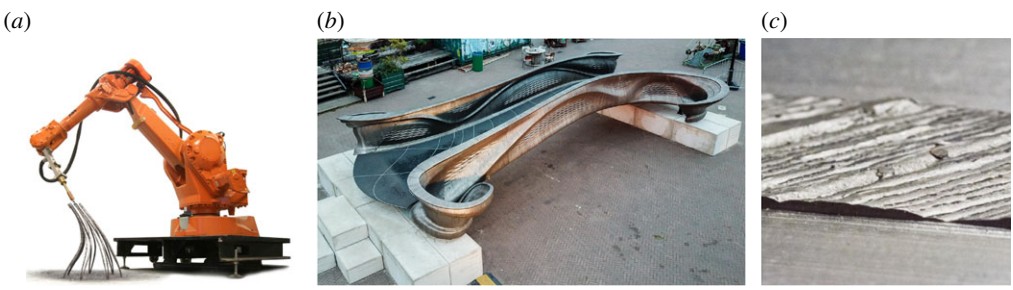

**Figure 1.** (*a*) The three-dimensional printing protocol developed by MX3D uses a weld head attached to a robotic arm (image by Joris Laarman, www.jorislaarman.com). (*b*) Pedestrian bridge manufactured using three-dimensional-printed steel [5]. (*c*) Close-up of the geometric variation on the surface of the material [6]. (Online version in colour.)

generally used factorial experimental design based on standard error regression and analysis of variance (or ANOVA) [15–19]. In parallel, extreme value theory has been deployed to analyse the effects of porosity [20–22]. At the level of individual components, it is necessary to take spatial dependence into account and for this, Gaussian random fields are most often used [23–25]. Surface roughness has received attention [26], with several papers investigating the use of compensation to allow creation of accurate shapes in the face of construction and discretization errors, though current approaches focus on two-dimensional sections [27–29]. Control charts for online surface quality control during the AM process have been developed [30,31]. Beyond AM, the field of *data-centric engineering* has seen explosive growth [32], and there is now an impetus to move away from traditional deterministic modelling approaches by, instead, creating models and methodologies which combine traditional modelling of engineering materials and structures with data science [33]. The role of machine learning in computational mechanics was reviewed by Oishi & Yagawa [34]; references therein describe how data-centric approaches have been developed for hyper-elastic materials, and extended to plasticity and other nonlinear or rate-dependent mechanics [35], including crystal plasticity, which is particularly relevant to AM with metallic material. An important distinction between previous work on AM and the present work is that our model is both generative *and* independent of any particular design, meaning that limited training data on one type of component may be sufficient to enable predictions for performance of another, as yet unseen component.

The paper is structured as follows: generative statistical models are constructed for the variable geometry and material properties of WAAM steel, respectively, in §§2 and 3. These models are validated in §4, where they are used to predict the behaviour of a yet-to-be-manufactured component. Conclusions, and implications for design and manufacture with WAAM steel, are contained in §5.

## 2. Statistical modelling of material geometry

The geometry of WAAM steel depends on factors that are not easily measured or controlled, motivating the treatment of material geometry as a random variable whose statistical properties can in principle be described. Statistical descriptions of rough material are well established [36], but these are mainly concerned with descriptive and regression-based methods [37–39] as opposed to formal generative modelling [40] (i.e. simulations of the random variable representing the geometry). In order to produce a finite-element-based prediction of the performance of a component before it is manufactured, a formal generative model for the upper $u^{(1)}$ and lower $u^{(2)}$ surface of the component is essential. In particular we note that this approach differs from traditional stochastic models of thin-walled structures [41,42] since the outer and inner face are neither assumed to be perfectly correlated (variable radius only) nor perfectly anti-correlated (variable thickness only).

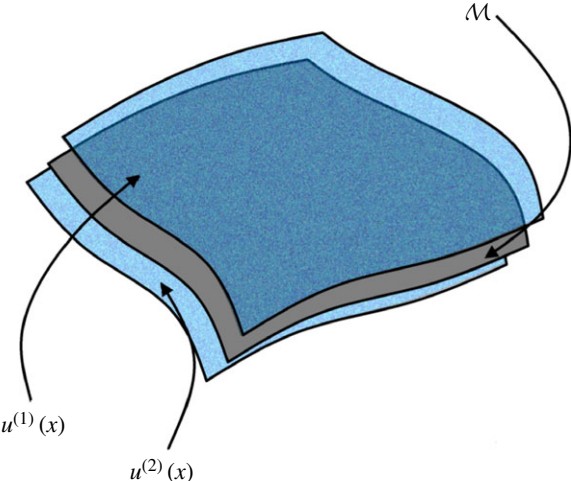

**Figure 2.** A model for geometric variation was constructed on a manifold $\mathcal{M}$, representing the notional centre of the component to be printed. This model consists of a bivariate stochastic partial differential equation (SPDE), whose solution is a vector field $u(x)$ defined on $x \in \mathcal{M}$, with coordinates $u^{(1)}(x)$ and $u^{(2)}(x)$ encoding the distance of the inside and outside faces from the manifold. (Online version in colour.)

A design for a WAAM steel component can be described by a two-dimensional manifold, $\mathcal{M} \subset \mathbb{R}^3$, representing the centre line of the component to be printed (figure 2). To build a generative model that is not specific to a particular choice of $\mathcal{M}$, we exploit the mathematical connection between Gaussian processes and stochastic partial differential equations (SPDEs), which decouple the statistical properties of the random field from the manifold $\mathcal{M}$ on which it is defined. Letting $u^{(1)}$ and $u^{(2)}$ denote the distance to the inside and outside faces, respectively, from $\mathcal{M}$, we consider the SPDE

$$\begin{bmatrix} \mathcal{L}^{(11)}(\mathbf{x}) & \mathcal{L}^{(12)}(\mathbf{x}) \\ \mathcal{L}^{(21)}(\mathbf{x}) & \mathcal{L}^{(22)}(\mathbf{x}) \end{bmatrix} \begin{bmatrix} u^{(1)}(\mathbf{x}) - m^{(1)} \\ u^{(2)}(\mathbf{x}) - m^{(2)} \end{bmatrix} = \begin{bmatrix} Z^{(1)}(\mathbf{x}) \\ Z^{(2)}(\mathbf{x}) \end{bmatrix}, \quad \mathbf{x} \in \mathcal{M}, \tag{2.1}$$

whose solution $[u^{(1)}, u^{(2)}]$ describes a stochastic realization of the design, where a class of models can be prescribed by different choices of the differential operators $\mathcal{L}^{(ij)}$ and the driving noise $Z^{(i)}$, $i, j \in \{1, 2\}$. In general, both the differential operators and the driving noise can depend on parameters, denoted $\phi$, which must be specified. The scalars $m^{(i)} > 0$ represent the mean displacement of surface $i \in \{1, 2\}$ from the manifold $\mathcal{M}$. For a discrete set of locations $\mathbf{x}_1, \ldots, \mathbf{x}_n \in \mathcal{M}$, the random vector $\mathbf{u}$ of length $2n$, whose entries are $u^{(i)}(\mathbf{x}_j)$, is Gaussian distributed with

$$p(\mathbf{u}|\phi) = \det \left( \frac{\mathbf{Q}_\phi}{2\pi} \right)^{1/2} \exp \left( -\frac{1}{2} \|\mathbf{Q}_\phi^{1/2}(\mathbf{u} - \mathbf{m})\|_2^2 \right), \tag{2.2}$$

where the precision matrix $\mathbf{Q}_\phi$ is determined by the choice of differential operators $\mathcal{L}^{(ij)}$ and the driving noise $Z^{(i)}$, $i, j \in \{1, 2\}$, while the mean vector $\mathbf{m}$ has entries $m^{(i)}$ in coordinates corresponding to $u^{(i)}$, $i \in \{1, 2\}$. Here, we make explicit the parameters $\phi$ of (2.1) that must be specified. Thus (2.2) represents a generative statistical model for the geometry of WAAM steel. The challenge is to identify suitable choices for the differential operators $\mathcal{L}^{(ij)}$, the driving noise $Z^{(i)}$, and the parameters $\phi$ that are most likely to have generated the training dataset $\mathbf{u}$, described next.

(*a*)

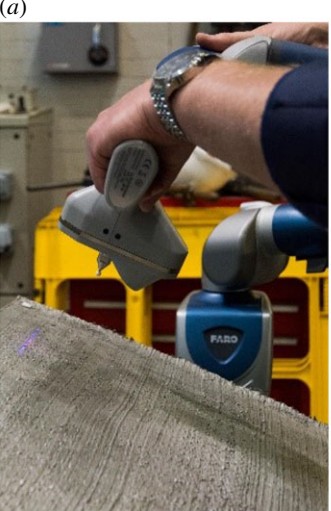

(*b*)

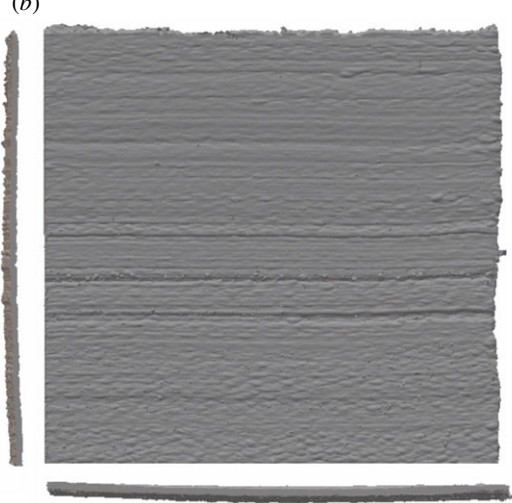

**Figure 3.** Laser scan of a three-dimensional-printed steel sheet. (*a*) Photograph of the hand-held scanning equipment. (*b*) Orthographic projection of (a portion of) the scanned sheet, denoted a panel. The notional thickness of the panel is 3.5 mm. (Observe a slight curvature in the notionally flat panel introduced by the residual stress.) (Online version in colour.)

## (a) Training dataset

To train our generative statistical model for WAAM steel, six notionally flat panels of approximate size $300 \times 300$ mm and a notional thickness of 3.5 mm were produced, using the experimental protocol described in appendix A(a), with one such panel displayed in figure 3. The actual geometry of the panel was recorded to 0.1 mm precision using a laser scanner. These data were stored as a point cloud, with $\mathbf{x} = x_1 \mathbf{e}_1 + x_2 \mathbf{e}_2 + x_3 \mathbf{e}_3$ used to denote an arbitrary point from the dataset as a point in $\mathbb{R}^3$. Using the data, convenient basis vectors $\mathbf{e}_1, \mathbf{e}_2, \mathbf{e}_3$ can be determined, with respect to which the coordinates $x_1, x_2, x_3$ are defined. To achieve this principal components of the point cloud dataset are computed, with the direction of least variation taken to be equal to the through-thickness direction $\mathbf{e}_1$. To align the panel in the $\mathbf{e}_2$ and $\mathbf{e}_3$ directions, a Gabor filter [43] is used to determine the direction of maximal alignment of the visible layer structure with $\mathbf{e}_2$. The final basis vector $\mathbf{e}_3$ is necessarily a unit vector orthogonal to $\mathbf{e}_1$ and $\mathbf{e}_2$ and represents the direction in which layers of steel are added. After centring the dataset, $x_1$ can be interpreted as the height of the surface of the panel, relative to the notional midpoint of the thickness, at the coordinate $(x_2, x_3) \in [0, 300]^2$. Thus the manifold $\mathcal{M}$, representing a panel in the training dataset, is $\mathcal{M} = \{(0, x_2, x_3) : 0 \leq x_2, x_3 \leq 300\}$. A regular $d \times d$ grid over $\mathcal{M}$, with $d = 300$, was used to determine the locations $\mathbf{x}_i, i = 1, \ldots, n, n = d^2$, at which the training dataset $\mathbf{u}$, appearing in (2.2), was constructed.

In order to model only those aspects of the geometry that are structurally relevant, two 'nuisance' aspects of the dataset need to be removed: (i) curvature due to residual stress in the panel, and (ii) weld splatter. The pre-processing steps taken to remove (i) and (ii) from the dataset are described in appendix A(b(i)).

## (b) Methodology for model selection

A set of 12 candidate models (denoted A, B, . . . , L) of the form in (2.1) and (2.2) was constructed, encoding combinations of geometric features that might be expected in WAAM steel, including anisotropy, non-stationarity, degrees of smoothness and oscillatory behaviour. These 12 models are defined by different choices for the differential operators $\mathcal{L}^{(ij)}$ and the driving noise $Z^{(i)}$,

with full details reserved for appendix A(b(ii)). To assess which combination of these geometric features best describes WAAM steel, we first fitted each model to the training dataset and then assessed goodness of fit in light of the complexity of each model. For training, maximum likelihood was employed due to its relatively low computational cost (e.g. compared to cross validation) and because it facilitates model selection under an Occam's razor principle, via the Akaike information criterion[1] (AIC) [44]

$$\text{AIC} = 2\,\dim(\hat{\phi}) - 2\log p(\mathbf{u}|\hat{\phi}), \tag{2.3}$$

where $\hat{\phi}$ denotes the values of the parameters

$$\hat{\phi} \in \arg\ \max_{\phi} \log p(\mathbf{u}|\phi), \tag{2.4}$$

that maximize the likelihood of observing the training dataset. The AIC balances goodness of fit with the complexity of the statistical model being fitted, with smaller AIC preferred. A difference in AIC between two models ($\Delta$AIC) of more than 10 implies that the model with the higher AIC has relatively little support [45].

The data likelihood in (2.2) requires a computational cost of $O(d^6)$ to evaluate exactly, where $d$ describes the resolution of the $d \times d$ grid on which data were obtained, due to the need to compute the determinant of the $d^2 \times d^2$ precision matrix $\mathbf{Q}_\phi$. This evidently becomes impractical for all but the crudest resolutions $d$ used to represent the dataset. To address this practical issue, a Markov random field approximation [46] of the stochastic process is adopted, which permits a sparse approximation to $\mathbf{Q}_\phi$ to be rapidly computed. The full details are contained in a separate technical report [47], and are identical to the finite-element approach set out in Lindgren *et al.* [46] which, in recent years, has become the standard approach to computation with spatially varying Gaussian random fields.

To numerically approximate the maximizer $\hat{\phi}$ of the likelihood in (2.4), we exploit an iterative numerical optimization method called 'natural gradient ascent' [48], that is

$$\phi^{(i+1)} = \phi^{(i)} + \alpha \mathbf{F}(\phi^{(i)})^{-1} \frac{\partial}{\partial \phi} \log p(\mathbf{u}|\phi^{(i)}),$$

where $\phi^{(0)}$ is an arbitrary initialization point, $\alpha > 0$ is a user-specified 'learning rate' taken to be 1, and $\mathbf{F}(\phi)$ is the Fisher information matrix with $(i,j)$th entry

$$\frac{1}{2}\text{tr}\left(\mathbf{Q}_\phi^{-1}\frac{\partial \mathbf{Q}_\phi}{\partial \phi_i}\mathbf{Q}_\phi^{-1}\frac{\partial \mathbf{Q}_\phi}{\partial \phi_j}\right).$$

The optimization algorithm terminates when the Euclidean norm of the natural gradient falls below a specified threshold. The computation of $\mathbf{F}(\phi)$ requires large dense matrices $\partial\mathbf{Q}_\phi/\partial\phi_i$ to be computed [49], which is impractical given the high resolution of the dataset aimed at. Therefore, to proceed, a surrogate likelihood $p(\tilde{\mathbf{u}}|\phi)$ based on the subset $\tilde{\mathbf{u}}$ of the dataset consisting of the central $50 \times 50\,\text{mm}$ portion of each panel is constructed. This surrogate likelihood is maximized using natural gradient ascent and the so-obtained value $\tilde{\phi}$ is substituted back into the exact likelihood when the AIC is computed. That is, $\hat{\phi}$ is replaced by $\tilde{\phi}$ in (2.3). This procedure can alternatively be motivated as a bespoke numerical optimization approach applied to the exact likelihood. Once $\tilde{\phi}$ has been found, further model-based computation can proceed with sparse precision matrices (only). In particular, an ensemble of synthetic geometries can be readily simulated.

## (c) Results

Figure 4*a* summarizes the model selection results, providing negative AIC values for each model in turn. The results provide strong statistical evidence for the role of orthotropic covariance structure ($\Delta\text{AIC} \geq 2 \times 10^4$ when comparing orthotropic models against the corresponding

---

[1]The use of AIC over alternative information criteria is a moot point in this work, due to the overwhelming information content in the dataset. That is to say, the use of alternative information criteria would not change our conclusion about the 'best' model.

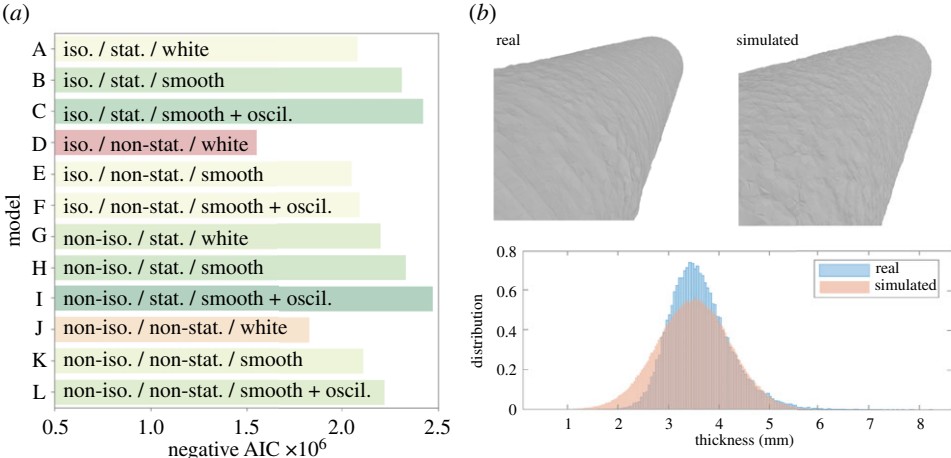

**Figure 4.** (*a*) Statistical model selection, based on the Akaike information criterion (AIC). (*b*) Examples of real and simulated circular hollow sections are shown on the top row, whilst on the bottom row, a comparison of wall thickness distributions is given. (Online version in colour.)

isotropic models; i.e. G versus A, H versus B, etc.), which is to be expected from the clearly visible banding structure on the material as steel is sequentially deposited. There is no support for spatially varying characteristic length scales perpendicular to the direction of the banding ($\Delta\text{AIC} \leq -2 \times 10^5$ when comparing the non-stationary models with spatially varying length scales against the corresponding stationary models; e.g. D versus A). Additionally, the data support smooth oscillatory driving noise (with globally constant periodicity) in the SPDE model ($\Delta\text{AIC} \geq 4 \times 10^4$ when comparing models with smooth oscillatory driving noise to those without oscillations or driven by white noise; e.g. C versus A and B).

The model with the lowest AIC (i.e. model I) is selected and can be used to generate synthetic geometries for general components, for example a circular hollow section (CHS) as shown in figure 4*b* (top), where its surface can be compared to the surface of a corresponding real component. These samples enable quantitative statistical predictions about geometric variation in a WAAM steel CHS to be produced. As an illustration, the wall thickness of a CHS component can be predicted from the ensemble of synthetic geometries; see figure 4*b* (bottom). The main limitation of our SPDE model is that it implies a Gaussian distribution for the wall thickness, whereas the real data displayed in figure 4*b* (bottom) is positively skewed. The reasons and implications for this are discussed below in the context of stochastic FEM simulations of the CHS.

## 3. Statistical modelling of material properties

To characterize the mechanical properties of WAAM steel independently from geometric variation, a large WAAM steel plate was manufactured using the experimental protocol described in appendix A(a), from which tensile coupons were cut and smoothed machined [50] at angles $\theta = 0°$, $45°$ and $90°$, where $\theta$ is defined such that layers of steel were added in the $90° - \theta$ direction under the printing protocol. Two coupons for each orientation were tested according to the EN ISO 6892-1 standard [51]. For each test, cross-sectional areas $A_c$ of each coupon were measured, while the testing machine measured the applied tensile load and a four-camera LaVision digital image correlation (DIC) system calculated averaged surface strain from both sides of the coupon.

Using these measurements, for each experiment, the longitudinal stress $\sigma_L$ at eight equally spaced longitudinal strain values $\varepsilon_L$ between 0% and 0.8% are recorded. Each experimental test is

modelled using a Ramberg–Osgood model [52],

$$\varepsilon_L = \frac{\sigma_L}{E_L(\theta)} + K\left(\frac{\sigma_L}{\sigma_0(\theta)}\right)^n, \tag{3.1}$$

where $E_L(\theta)$ and $\sigma_0(\theta)$ are the unknown longitudinal elastic modulus and yield strengths, respectively, for a coupon with layers oriented at an angle $\theta$, $n$ is an unknown (scalar) hardening exponent and $K$ is a constant. Here, $K$ is taken equal to 0.002 and, as a consequence, the yield strength $\sigma_0$ corresponds to the 0.2% proof stress, which is widely used to define the yield of metals in the literature [50].

Collectively, by testing coupons over a range of angles, the Ramberg–Osgood model can be used to define a general anisotropic homogeneous, elasto-plastic constitutive law under plane stress conditions over all angles $\theta$. In this contribution, the elastic part of the deformation is consider to be transversely isotropic, parameterized by $E_1, E_2, G_{12}$ and $v_{12}$, respectively, the elastic moduli perpendicular and parallel to the build direction, a shear modulus and a Poisson ratio. For the plastic response, Hill's quadratic yield surface [53] is considered which, under plane stress assumptions, is uniquely parameterized by three parameters $(F, G, N)$. Post-yield, a simple isotropic hardening rule is defined by the single exponent, $n$. The mathematical connection between the Ramberg–Osgood model and the two-dimensional elasto-plastic material parameters used in the structural scale calculations are provided in the appendix.

## (a) Bayesian inference and computation

To infer the distribution of material parameters a Bayesian approach is used, such that a generative model is obtained. All nine unknown parameters are collected into a single vector

$$\boldsymbol{\xi} = [E_1, E_2, G_{12}, v_{12}, \sigma_y(0°), \sigma_y(90°), \sigma_y(45°), n, \sigma_e].$$

The data are represented in a vector $\mathbf{d} \in \mathbb{R}^{48}$, which contains eight evenly spaced strain values for each of the six experiments from 0.1% up to 0.8% strain at given recorded values of stress. The mapping between parameter values and data is defined by the forward map $\mathcal{G}(\boldsymbol{\xi}) : \mathbb{R}^9 \to \mathbb{R}^{48}$, and permits the definition of a stochastic model

$$\mathbf{d} = \mathcal{G}(\boldsymbol{\xi}) + \mathbf{e}.$$

Here, $\mathbf{e}$ is a random vector that represents measurement error, whose components are independent and Gaussian with $e_i \sim N(0, s_e^2 d_i^2)$, whereby the noise-to-signal parameter $s_e$ is to be learned, providing a quantitative measurement of uncertainty in both the data and the model (i.e. allowing for mis-specification). Weakly informative prior distributions (informed by [6]) are taken for each of the material parameters, with mean values

$$E_1 = 154.0\,\text{GPa}, \quad E_2 = 148.0\,\text{GPa}, \quad G_{12} = 222.6\,\text{GPa} \quad \text{and} \quad v = 0.3,$$

for the elastic properties, and

$$\sigma_0(0°) = 0.36\,\text{GPa}, \quad \sigma_0(90°) = 0.325\,\text{GPa}, \quad \sigma_0(45°) = 0.405\,\text{GPa} \quad \text{and} \quad n = 12.9$$

for the plastic parameters. The posterior is numerically approximated using a Markov chain Monte Carlo (MCMC) method; in particular, an adaptive Metropolis–Hastings algorithm [54], implemented as a customized model in PyMC3.[2] To obtain convergence diagnostics, four independent chains were run for 24 000 iterations; all parameters obtained $\hat{R}$ values of less than 1.02, suggesting the Markov chain had converged. The first 1000 iterations (from each chain) were discarded as burn-in to eliminate the bias imposed by the starting positions of the chains. The results of the MCMC simulations are summarized in the main text, while marginal posterior distributions of parameters (alongside their priors) are given in figure 5.

---

[2]See https://docs.pymc.io.

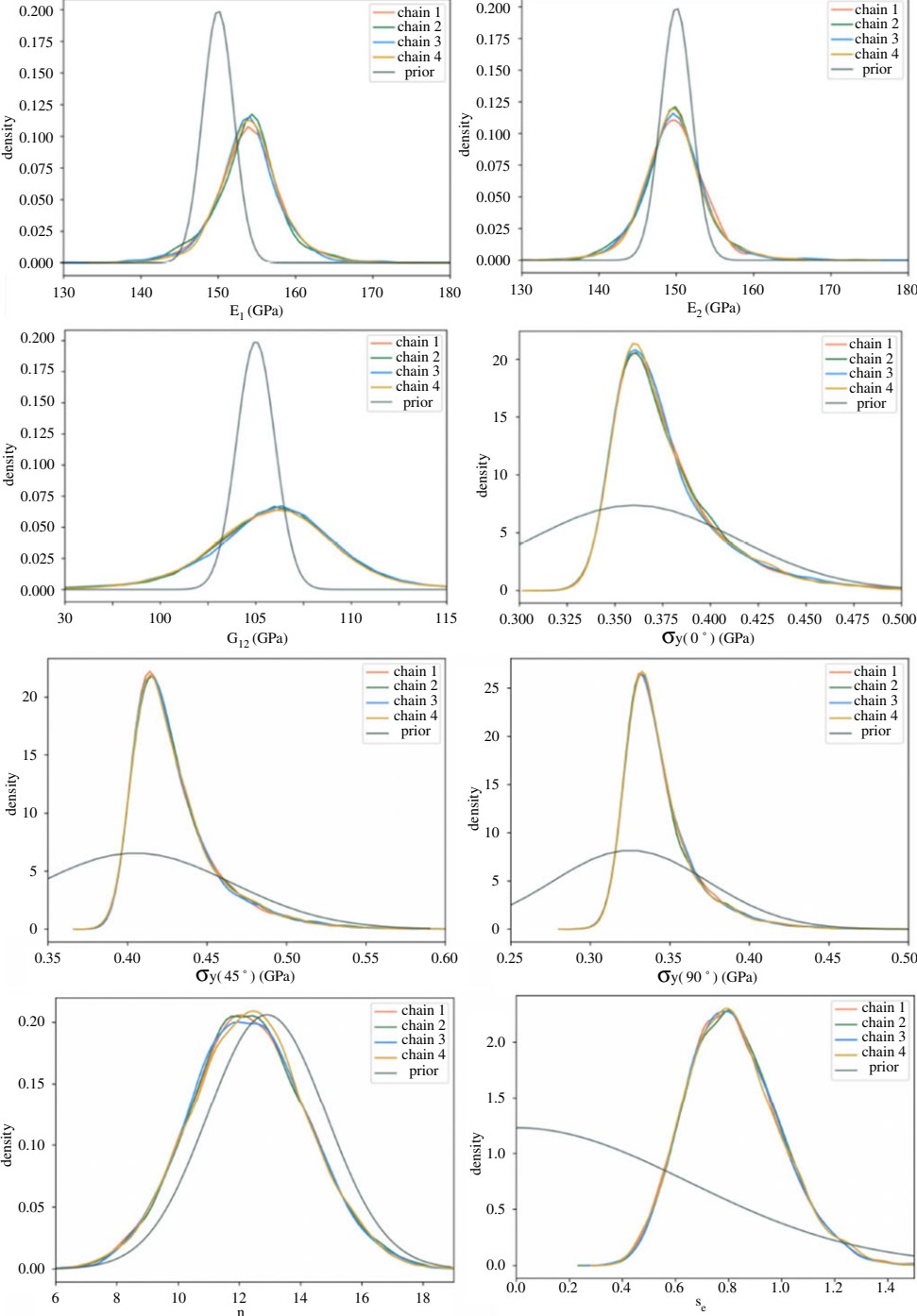

**Figure 5.** Marginal distributions for each of eight material parameters, generated by four independent chains using an Adaptive Metropolis–Hastings in `pymc3`, shown alongside prior distributions. The distribution of $\nu$ is not included since there is no observable difference from the prior, as might be expected due to the uniaxial nature of the tests. (Online version in colour.)

## (b) Inferred material properties

Figure 6 provides a summary of the outputs of the MCMC computations, and table 1 gives the posterior mean values and marginal standard deviations for each parameter. Figure 6*a* displays

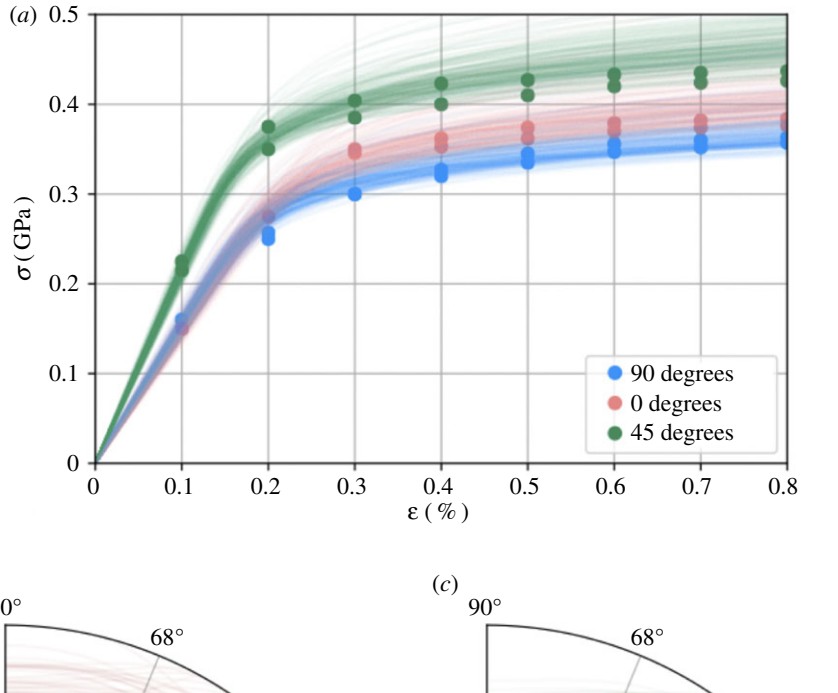

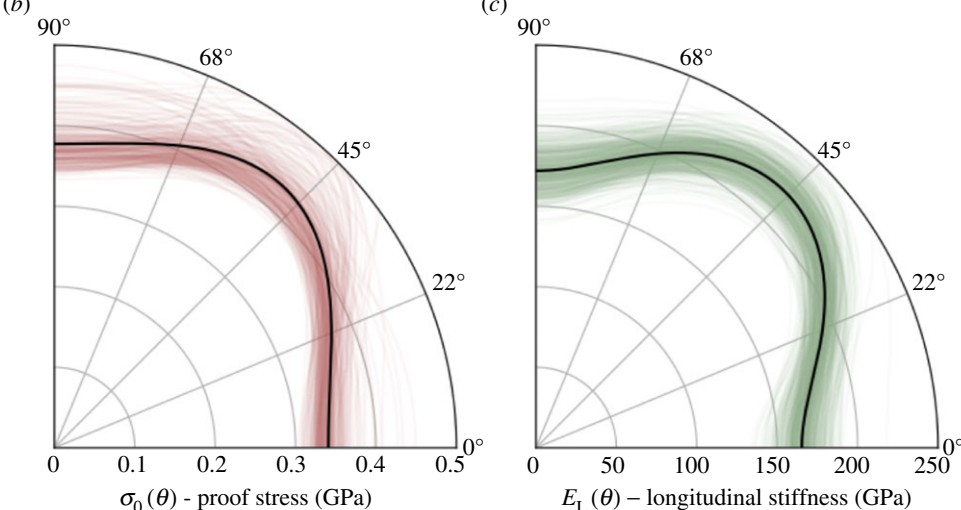

**Figure 6.** Bayesian inference for parameters of the Ramberg–Osgood model. (*a*) Posterior realizations of stress–strain curves against experimental data displayed as dots. (*b,c*) Posterior samples for longitudinal stiffness (*c*) and proof stress (*b*) as a function of build angle $\theta$. Bold lines show maximum *a posteriori* estimates. (Online version in colour.)

posterior samples, showing the implied distribution over stress–strain curves in relation to the training dataset (solid dots). The two polar plots in figure 6*b,c* show posterior samples (green and red) as well as the maximum *a posteriori* estimate of longitudinal elastic modulus and proof stress over the sweep of possible build angles $\theta$. Isotropic behaviour would be observed as a quarter circle (constant radius) for both properties. For both elastic and yield properties, the posterior samples clearly show an anisotropic material response.

The elastic anisotropy is a consequence of both the strong texture of the material (preferential alignment of the crystals), and the manifestation of strong single crystal elastic (and plastic) anisotropy of Fe at the macroscale.

## (i) Strong material texture of WAAM

During the build, cubic metals like stainless steel grow epitaxially with the ⟨100⟩ type directions tracking the direction of the highest thermal gradient during deposition, i.e. the build direction

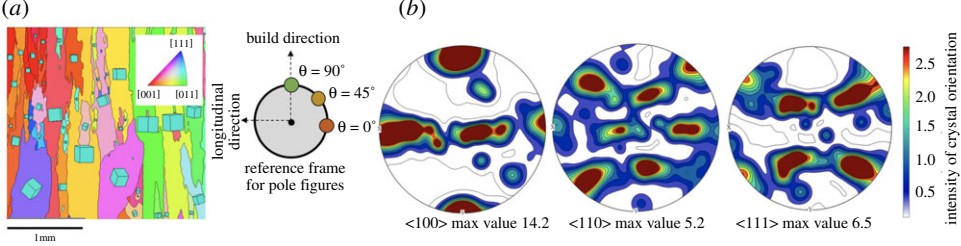

**Figure 7.** Material analysis. (*a*) Inverted pole figure map of an examined surface, with build direction running vertically. (*b*) Pole figures of the examined surface, showing strong texture, with preferential alignment of the crystals, within WAAM steel. (Online version in colour.)

**Table 1.** Posterior summary statistics for parameters of the Ramberg–Osgood model, (A 3).

| parameter (units) | mean | s.d. | 3–97% interval |
|---|---|---|---|
| $E_1$ (GPa) | 154 | 8.42 | 139–170 |
| $E_2$ (GPa) | 151 | 8.72 | 136–168 |
| $G_{12}$ (GPa) | 108 | 7.95 | 93–123 |
| $\nu_{12}$ | 0.286 | 0.02 | 0.249–0.323 |
| $\sigma_0(0°)$ (MPa) | 343 | 22.3 | 311–387 |
| $\sigma_0(90°)$ (MPa) | 376 | 28.1 | 337–433 |
| $\sigma_0(45°)$ (MPa) | 429 | 27.7 | 391–482 |
| $n$ | 12.3 | 1.90 | 8.78–15.96 |
| $s_e$ (%) | 0.82 | 0.18 | 0.5–1.15 |

[55]. This is supported by figure 7*a* which shows the inverted pole figure map of a specimen surface overlayed with cubic unit cells representing each grain orientation [50]. Long and columnar grains are observed, with a preferential orientation along the build axis. The strong texture can also be observed in ⟨100⟩, ⟨110⟩ and ⟨111⟩ pole figures (figure 7*b*), obtained through conducted electron backscatter diffraction analysis (or EBSD analysis). The contours denote the intensity of specific crystal orientation distributions that were observed.

## (ii) Single crystal anisotropy

Austenitic steels exhibit strong single crystal anisotropy due to their face-centred cubic crystal structure. This structure forms when a sufficient quantity of nickel (10%) is added to a 20% chromium alloy. As a result, the elastic modulus along the build direction (90°), and the orthogonal direction (0°) are approximately 155 GPa. At 45° to ⟨100⟩, the elastic modulus is much higher, approximately 215 GPa, leading to a mean shear modulus $G_{12} \sim 108$ GPa. This anisotropy extends into full three-dimensional anisotropy, with the ⟨111⟩ being the stiffness direction [56]. Face-centred cubic Fe is also plastically anisotropic. The material exhibits the strongest directions at ⟨111⟩ and ⟨110⟩ to the crystal directions. Both directions are near 45° to ⟨100⟩, supporting the posterior estimates obtained with a mean proof stress for 45° coupons 429 MPa, while for 0° and 90° the lesser values of 343 MPa and 376 MPa, respectively, were obtained (table 1).

The inclusion of an unknown noise-to-signal parameter $s_e$ in the likelihood helps to represent uncertainty introduced not only by the fact that the experimental measurements are noisy, but also because all models will, to some extent, be mis-specified. A posterior mean of $s_e = 0.82\%$ is obtained, indicating a reasonable fit. It can be observed that $s_e$ is most strongly correlated with

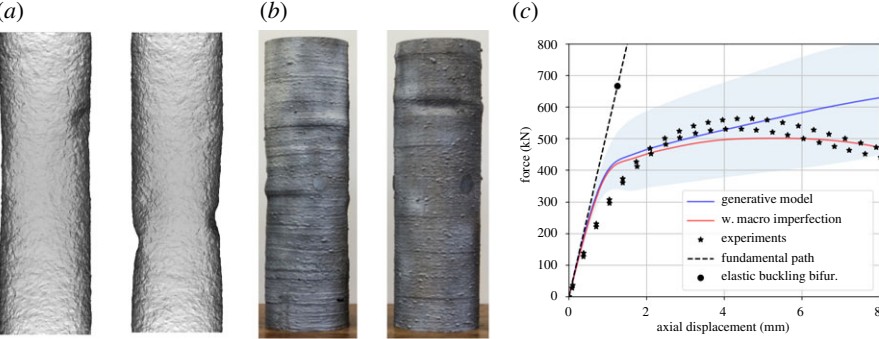

**Figure 8.** (*a*) Samples from the trained generative model. (*b*) Buckled cylinders, experimentally produced [6]. (*c*) Predicted and experimental load-displacement curves based on the generative model, with and without macro-scale geometric imperfections introduced, and a perfect elastic model. Two s.d. regions are shaded. (Online version in colour.)

proof stress values $\sigma_0(\theta)$, suggesting that the uncertainty in the model predictions is mostly driven by the yield values.

# 4. Uncertainty propagation up the structural scales

The generative statistical models for geometric and mechanical variation in WAAM steel are combined into a unified generative statistical model that treats these two sources of variation as independent. However, it is *a priori* unclear whether these two sources of variation are approximately independent or not, and the implication of this assumption will be discussed in view of the results. To assess the predictive performance of this unified generative model, we consider compressive cross-section testing of a CHS (figure 8). The axial compression of thin-walled CHS is known to exhibit complicated stability behaviour [57]; this therefore provides a realistic and challenging test for our generative model. In particular, no experimental data from a CHS are used—all parameters are inferred from simpler tests at a lower structural length scale.

## (a) Stochastic finite-element model of circular hollow section

The statistical models described above are combined into a unified statistical model whose instantiations can be realized in the finite-element software ABAQUS [58], in order that the nominal design of a CHS can be endowed with realistic surface geometry and material properties according to this unified statistical model.

A high-resolution finite-element model (FEM) of the CHS is constructed in ABAQUS [58], using approximately 200 k quadratic, tetrahedral finite elements. The CHS has diameter 170 mm, length 580 mm and notional thickness 3.5 mm. The output generates a load-displacement curve as shown in figure 8*c*. Coupling the FEM to our unified generative model for geometric and mechanical variation gives rise to a stochastic FEM model. As a baseline, an eigenvalue analysis of the buckling perfect elastic system is also carried out.

To instantiate a sample from the unified statistical model, a notional CHS is first constructed as a fine mesh in ABAQUS. From the set of nodal coordinates, nodes $\mathbf{x}_i^{(1)}$ on the outer surface and nodes $\mathbf{x}_i^{(2)}$ on the inner surface of the CHS are identified. The SPDE description of the fitted geometry model is used to generate a solution $\mathbf{u}(\mathbf{x}_i^{(j)})$ of the SPDE defined at each of the nodes $\mathbf{x}_i^{(j)}$. The first component, $u_1(\mathbf{x}_i^{(1)})$, describes the amount that $\mathbf{x}_i^{(1)}$ should be moved in the outward normal direction $\mathbf{n}(\mathbf{x}_i^{(1)})$ to the nominal CHS, producing a displaced node $\mathbf{y}_i^{(1)}$. Similarly, the second component, $u_2(\mathbf{x}_i^{(2)})$, describes the amount that $\mathbf{x}_i^{(2)}$ should be moved in the inward normal

direction $-\mathbf{n}(\mathbf{x}_i^{(2)})$ to a new node position $\mathbf{y}_i^{(2)}$, i.e.

$$\mathbf{y}_i^{(1)} = \mathbf{x}_i^{(1)} + u_1(\mathbf{x}_i^{(1)})\mathbf{n}(\mathbf{x}_i^{(1)}) \quad \text{and} \quad \mathbf{y}_i^{(2)} = \mathbf{x}_i^{(1)} - u_2(\mathbf{x}_i^{(2)})\mathbf{n}(\mathbf{x}_i^{(2)}).$$

In the ABAQUS input file, the original nodal coordinates $\mathbf{x}_i^{(j)}$ are replaced with the deformed coordinates $\mathbf{y}_i^{(j)}$ generating a mesh of an imperfect CHS, with the original triangulation. Finally, interior nodes in the fine mesh are re-positioned along the radial direction such that the interior angles of the elements are approximately preserved.

In practice, to control the dimension of the Gauss–Markov approximation to the SPDE, an intermediate step is included, wherein the SPDE is solved on a coarser mesh and the values $u_j(\mathbf{x}_i^{(j)})$ of the solution on the fine mesh are obtained by interpolation from the coarser mesh. Interpolation introduces a small amount of additional smoothing compared to the original solution of the SPDE, but the resolution of the coarse mesh was taken to be 1 mm, which is expected to render the impact of additional smoothing negligible at the level of the salient geometric variation of WAAM steel.

ABAQUS provides a basic interface to define anisotropic elasto-plastic material models defined by Hill's criterion, as used in the material model above. Sampled random material parameters, randomly drawn for the set of posterior samples generated in §3, can simply be input into this FEM's material model. In the experiment described, these are either constant for the whole cylinder, or locally defined on a single element. For all computations, both ends of the column are tied to a central reference point in their respective planes, and the boundary conditions are applied to those reference points. The bottom of the column is defined to be *encastré* (all degrees of freedom constrained to zero), whilst at the top of the cylinder all degrees of freedom are constrained to zero, except the axial displacement (direction 3). A quasi-static Riks analysis is used to solve the problem up to a compressive axial displacement of 10 mm. From the solution outputs at each loading step, load-displacement curves can be generated.

## (b) Axial compression test results

The stochastic FEM produces load-displacement curves representing predictions for how a CHS deforms under an axial compression test. The results are summarized (in blue) in figure 8c. Figure 8a depicts two such realizations of the CHS (at the same compressive displacement, $\Delta = 4.0$ mm). Note that, due to the random geometric imperfections, post-buckled 'dimples' (not observed before loading) develop in different positions for each realization sampled. To assess the quality of these statistical predictions, two physical tests of CHS were carried out [5] and the experimental samples are depicted in figure 8b. The distribution of load-displacement curves is roughly consistent with these two experimental tests. Since our generative model has been trained only on limited data from flat panels and coupons, with no training data on CHS used, these ball-park results are highly encouraging in the context of predicting the behaviour of components that are yet to be manufactured. Moreover, the predicted deformed geometry in figure 8a is characteristically similar to that observed in the experiments (figure 8b), with the clear development of dimples at positions of localized plastic buckling.

The main differences in the distribution of simulated outputs compared with the experiments is seen in (i) the initial stiffness and (ii) transition to failure. For the simulation, the load-displacement response shows a sharp change in stiffness at the onset of plastic failure. In the experiment, this transition is more gradual, typical of a buckling response with a more significant geometric imperfection [57]. This is made apparent when the experimental curves are compared to the fundamental path and buckling bifurcation points calculated for the perfect elastic system, as shown by the dashed black line and circle in figure 8c.

Importantly, our generative model focuses on *local* geometric variation (only) in WAAM, while larger imperfections introduced during manufacture of the CHS are not modelled. To account for their effect, additional data on such macro-scale imperfections would be required. However, as a

preliminary test to determine the importance of macro-scale imperfections, we apply the standard engineering approach of adding a geometric imperfection in the form of the eigenmode associated with the smallest computed eigenvalue of a 'perfect' elastic CHS. Guided by preliminary measurements in Gardner *et al.* [5], a maximum imperfection amplitude of 2 mm is added to the initial CHS geometry and the resulting load-displacement response is shown as the red curve in figure 8*c*. The inclusion of a such an imperfection provides a pronounced de-stiffening effect due to a localized buckling response, with better agreement at higher compressive displacements ($\Delta = 6$–8 mm) compared to the generative model. Finally, we highlight that our generative model does not attempt to capture uncertainties introduced by the experimental test itself, such as the uncertainties associated with true experimental boundary constraints and uniformity of loading.

## 5. Conclusion: implications for design with three-dimensional-printed steel

Approximate predictions for the performance of a hypothetical component are a prerequisite for efficient and low-cost design using WAAM. Our results demonstrate that relatively small amounts of training data, gathered from tests at centimeter scales, can be exploited to train a generative statistical model for AM steel, in turn enabling approximate predictions of performance at higher structural length scales to be produced. Regarding safety-critical certification, however, experimental testing at higher length scales remains imperative, since predictive uncertainties at these length scales are still substantial (as evidenced by the predictive uncertainty in figure 8*c*).

This study paves the way for future research exploiting stochastic simulation to identify which aspects of performance at higher length scales are most uncertain, and to target more expensive, large-scale tests in such a way that these remaining sources of uncertainty can be minimized. Our generative model will be well-suited to this task, enabling the quantification of (and distinguishing between) both epistemic uncertainty, due to the limited training data, and aleatoric uncertainty, due to variation inherent to the printing protocol. Stochastic computational pipelines are set to play a key role in minimizing the cost of certification of new structures into service, given that even minor changes to printing protocols have the potential to change the nature of variation in material properties on all structural scales, introducing additional uncertainty that will need to be quantified.

Data accessibility. This article has no additional data.

Authors' contributions. T.D., L.G., M.A.G., C.J.O. devised the original research project. C.B., P.K., L.G. performed the experiments that constituted the dataset. L.F., W.S.K., C.J.O. developed the geometric model. T.D., G.D., R.S. developed the material model and analysis. T.D., L.F., P.G. developed the finite-element model. T.D., C.B., R.S., W.S.K., M.A.G., C.J.O. co-wrote the manuscript.

Competing interests. We declare we have no competing interests.

Funding. This research was supported by the Lloyd's Register Foundation programme on data-centric engineering at the Alan Turing Institute, UK. Material samples were provided by MX3D. This research benefited from EPSRC and UKRI funding under grant nos. EP/R010161/1, EP/R017727/1, EP/K031066/1 and 2TAFFP/100007 and made use of the Rocket HPC service at Newcastle University.

Acknowledgements. T.D. wishes to thank Prof. Joao Quinta da Fonseca and Dr Pratheek Shanthraj (both University of Manchester, UK) for informative discussions and suggestions regarding the anisotropy and texture of WAAM steel.

## Appendix A

### (a) Experimental protocol

For building the components that comprise the training dataset, wire of 1.0 mm diameter was used while the employed welding speed and wire feed rate were 15–30 mm s$^{-1}$ and 4–8 m min$^{-1}$,

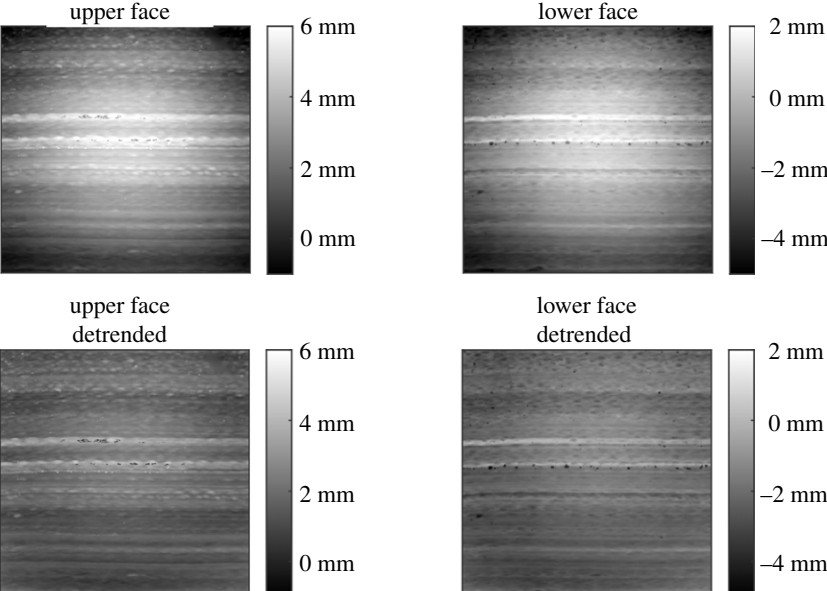

**Figure 9.** Digital removal of residual stress. The top row displays unprocessed upper and lower surface heights around a notional flat plane in $\mathbb{R}^3$, for one panel in the dataset. Residual stress in the panel manifests itself as global curvature and is most pronounced in the corner regions. The bottom row displays the same data after a quadratic trend was removed.

respectively. The employed shield gas was 98% AR and 2% $CO_2$ at a flow rate of 10–20 l min$^{-1}$, the current and arc voltage of the deposition process were 100–140 A and 18–21 V, respectively, while the deposition rate was between 0.5 and 2.0 kg h$^{-1}$.

## (b) Statistical modelling of material geometry

This appendix first describes how the data are pre-processed, then presents the 12 statistical models that were considered.

### (i) Data pre-processing

In order to model only those aspects of the geometry that are structurally relevant, two 'nuisance' aspects of the dataset need to be removed: (i) curvature due to residual stress in the panel, and (ii) weld splatter.

The first issue arises because each panel was cut from a larger sheet of material. Consequently, the residual stresses due to cooling of the material in the larger sheet are taken out of equilibrium when the panel is removed. This causes the panel to be non-flat at equilibrium on the macro scale, and manifests itself as a slight global curvature in the $\mathbf{e}_1$ direction of the panel. As our focus was not on residual stress, a statistical pre-processing technique is used to detrend each panel dataset. A quadratic surface $(x_1, x_2, x_3) \mapsto (s(x_2, x_3), x_2, x_3)$ is fitted to the full collection of points in the laser scan dataset, and the fitted linear and quadratic terms $s(x_2, x_3)$ are then subtracted from every point; $(x_1, x_2, x_3) \mapsto (x_1 - s(x_2, x_3), x_2, x_3)$; see figure 9. Although a quadratic fit leaves some remaining global curvature in the dataset, it is considered to have negligible impact on the subsequent analyses that we report.

The second issue occurs due to splattering of molten steel on lower layers during the application of subsequent layers of material, and manifests itself as small beads of steel affixed to the surface of the panel. Splatter is assumed to have negligible effect on engineering performance of the material. Accordingly, to circumvent the need to specify a generative statistical model for

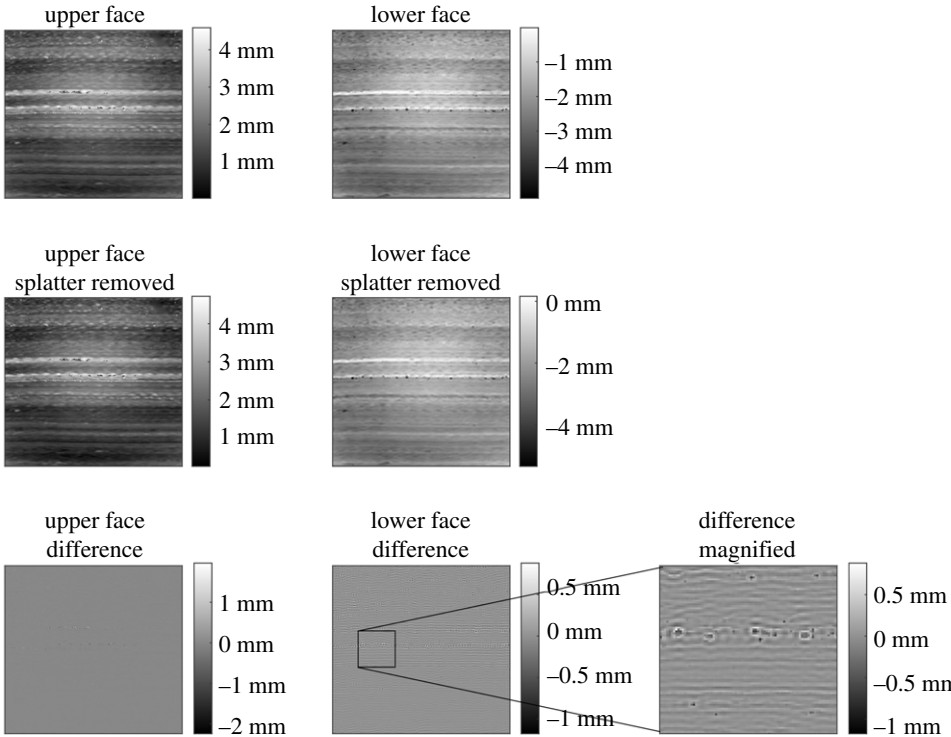

**Figure 10.** Digital removal of weld splatter. The top row displays (detrended) upper and lower surface heights for one panel in the dataset. The middle row displays the same data after weld splatter was removed. The bottom row displays the difference between the top and middle rows, i.e. the splatter pattern that is being removed.

splatter, the dataset was digitally processed to remove instances of splatter. This was achieved by noting that geometric variation due to splatter is characterized by higher-frequency information, due to the localized nature of the splatter, such that it can be removed using a truncation of the Fourier transform. Let $\mathbf{g}_{i,j} = (g_i, g_j)$ denote the $(i,j)$th element of a uniform $d \times d$ grid over the region $[0, 300]^2$ of the $\mathbf{e}_2$-$\mathbf{e}_3$ plane and let $H^{(1)}$ and $H^{(2)}$ denote matrices whose $(i,j)$th elements represent, respectively, the actual height of the front and rear faces of the panel, based on the (quadratic detrended) laser scan dataset, at $\mathbf{g}_{i,j} \in [0, 300]^2$. The entries of $H^{(1)}$ and $H^{(2)}$ are computed by interpolating nearest neighbour data from the point cloud. A 1 mm resolution is used when discretizing the dataset, to capture the salient geometric variation in the material whilst controlling subsequent computational cost. Let $\hat{H} = \mathcal{F}(H)$ denote the two-dimensional Fourier transform of a matrix $H$. Our pre-processing step, denoted $\hat{H} \mapsto \mathcal{T}(\hat{H})$, sets to zero those entries of $\hat{H}$ corresponding to high-frequency modes $\omega = (\omega_i, \omega_j)$ with $i$ or $j$ larger than $I_{\max}$. Application of the inverse fast Fourier transform leads to a matrix $\mathcal{F}^{-1}(\mathcal{T}\hat{H})$ which approximates the height of the face of the panel with splatter removed. The overall procedure $H \mapsto (\mathcal{F}^{-1} \circ \mathcal{T} \circ \mathcal{F})(H)$ is applied to both $H^{(1)}$ and $H^{(2)}$, leading to matrices that are denoted $\tilde{H}^{(1)}$ and $\tilde{H}^{(2)}$. Here, $I_{\max}$ is manually selected to provide a good compromise between removal of splatter and avoidance of smoothing of the relevant variation in the dataset (figure 10).

### (ii) Candidate models

To capture the salient features of the dataset a selection of candidate models is constructed, each model based on a Gaussian process and characterized by its precision matrix $\mathbf{Q}_\phi$. To this end, we exploited the relationship between Gaussian processes and SPDEs [46]. In particular,

we considered a coupled system of SPDEs [59,60] of the form in (2.1), together with Neumann boundary conditions on the boundary of $\mathcal{M}$, where $\mathcal{L}^{(ij)}$ are differential operators to be specified and $Z^{(i)}$ are independent stochastic processes driving the system of SPDEs, also to be specified. The parameter $\phi$ represents any degrees of freedom that must be specified, either in the operators $\mathcal{L}^{(r,s)}$ or in the noise process $Z^{(r)}$. The candidate models are now described.

## Isotropic stationary model $\mathcal{L}^{(ij)}$ (used in models A, B, C)

Our simplest model has the form

$$\mathcal{L}^{(ij)}u^{(j)}(\mathbf{x}) = (\eta^{(ij)} - \Delta)(\tau^{(ij)}u^{(j)}(\mathbf{x})),$$

where $\Delta$ denotes the Laplacian on $\mathcal{M}$. Here, $\eta^{(ij)}, \tau^{(ij)} > 0$, $(\eta^{(ij)})^{-1/2}$ can be viewed as a length-scale parameter for $u^{(j)}$ and $(\tau^{(ij)})^{-1}$ can be viewed as a scale parameter controlling the magnitude of the values of $u^{(j)}$. It is assumed that $\eta^{(12)} = \eta^{(21)}$, $\tau^{(12)} = \tau^{(21)}$, $\eta^{(11)} = \eta^{(22)}$ and $\tau^{(11)} = \tau^{(22)}$, reflecting the hypothesis that the statistical properties of the front and rear faces are expected to be similar. There are thus 4 d.f. in the differential operators $\mathcal{L}^{(ij)}$ to be estimated: $\phi = \{\tau^{(11)}, \tau^{(12)}, \eta^{(11)}, \eta^{(12)}\}$.

The role of the Laplacian, as a second-order differential operator, reflects the hypothesis that the surface is somewhat rough, typically having first derivatives that are mean square integrable and no higher derivatives when the driving noise $Z^{(i)}$ is standard white noise. (Models with additional smoothness are considered in the sequel.)

## Anisotropic stationary model $\mathcal{L}^{(ij)}$ (used in models G, H, I)

The WAAM process introduces clear banding, which indicates that an isotropic model may be inappropriate to describe the geometric variation in the material. The simplest non-isotropic model considered is the orthotropic stochastic process model, which takes the form

$$\mathcal{L}^{(ij)}u^{(j)}(\mathbf{x}) = (\eta^{(ij)} - \nabla \cdot \mathbf{H}^{(ij)}\nabla)(\tau^{(ij)}u^{(j)}(\mathbf{x})),$$

for a diagonal positive definite matrix $\mathbf{H}^{(ij)}$ to be specified. The model is again driven by a standard white noise process $Z^{(i)}$ and $\mathbf{H}^{(ij)} = \mathbf{H}$ for a fixed matrix $\mathbf{H} = \mathrm{diag}(h_1, h_2, h_3)$ for all $ij \in \{1, 2\}$, reflecting the assumption that the statistical properties of the front and the rear surface are considered to be identical. Furthermore, $h_1$ and $h_2$ are assumed to be equal, to reflect the fact that a general component will have no preferred orientation in the $\mathbf{e}_1$-$\mathbf{e}_2$ plane, but the $\mathbf{e}_3$ direction is distinguished as the direction in which the layers are added.

## Non-stationary model $\mathcal{L}^{(ij)}$ (used in models D, E, F, J, K, L)

To allow for the possibility that the statistical properties of the material geometry are non-stationary, the introduction of spatially dependent coefficients $\eta^{(ij)}$, $\tau^{(ij)}$ into the differential operator is also considered. The isotropic model, for example, becomes

$$\mathcal{L}^{(ij)}u^{(j)}(\mathbf{x}) = (\eta^{(ij)}(\mathbf{x}) - \Delta)(\tau^{(ij)}(\mathbf{x})u^{(j)}(\mathbf{x})),$$

and the anisotropic model is similarly modified. It is not clear how one can usefully parametrize functions $\eta^{(ij)}(\cdot)$ and $\tau^{(ij)}(\cdot)$ in general, since any parametrization will be specific to the coordinate system of $\mathcal{M}$ in which the data are defined. However, recognizing that the $\mathbf{e}_3$ direction is distinguished as the direction perpendicular to the weld bands, it is possible to consider functions of the form $\mathbf{x} \mapsto \eta^{(ij)}(x_3)$ and $\tau^{(ij)}(x_3)$ only, and still retain the ability to generalize across different manifolds, representing different components that could be produced. For this paper, we considered a representation of these functions of the form

$$\eta^{(ij)}(x_3) = c^\eta \left(1 + \sum_{k=1}^{4}\sum_{l=0}^{1} \gamma_{k,l}^\eta \sin\left(\frac{2\pi k}{100}x_3 + \frac{l\pi}{4}\right)\right),$$

and similarly for $\tau^{(ij)}$. Thus we must estimate in addition the parameters $c^\eta$, $c^\tau$, $\gamma_{k,l}^\eta$ and $\gamma_{k,l}^\tau$, $k \in \{1, 2, 3, 4\}$, $l \in \{0, 1\}$.

### White noise model $Z^{(i)}$ (used in models A, D, G, J)

Here, each $Z^{(r)}$ is an independent standard white noise process on $\mathcal{M}$ (i.e. a generalized random field).

### Smoother noise model $Z^{(i)}$ (used in models B, E, H, K)

Replacing the driving noise process $Z^{(r)}$ by a smoother process leads to a smoother model for geometric variation in the material. The smoother model that we considered for the driving noise $Z^{(r)}$ takes the form

$$(\eta^{(r)} - \nabla \cdot \mathbf{H}\nabla)Z^{(r)}(\mathbf{x}) = W^{(r)}(\mathbf{x}), \tag{A 1}$$

where $W^{(r)}$ is a standard white noise process on $\mathcal{M}$ and this SPDE is equipped with Neumann boundary conditions on the boundary of $\mathcal{M}$. In the isotropic case, we take $\mathbf{H} = \mathbf{I}$, otherwise we take $\mathbf{H} = \text{diag}(h_1, h_1, h_3)$ as previously described. The use of smoother noise introduces an additional parameter $\eta^{(r)} > 0$ to be estimated. Note that, by substituting equation (A 1) into equation (2.1), these smoother models can be viewed as a fourth-order SPDE driven by white noise [46].

### Smooth oscillatory noise model $Z^{(i)}$ (used in models C, F, I, L)

Finally, the smoother model is extended to include in addition an oscillatory component. This is achieved [46] by replacing equation (A1) with

$$(\eta e^{i\pi\vartheta} - \nabla \cdot \mathbf{H}\nabla)[Z^{(1)}(\mathbf{x}) + iZ^{(2)}(\mathbf{x})] = W^{(1)}(\mathbf{x}) + iW^{(2)}(\mathbf{x}), \tag{A 2}$$

where again $W^{(r)}$ is a standard white noise process on $\mathcal{M}$ and this SPDE is equipped with Neumann boundary conditions on the boundary of $\mathcal{M}$. The real and imaginary parts, $Z^{(1)}$ and $Z^{(2)}$, of equation (A 2) can be shown to be independent and can be used as the driving noise processes in the SPDE. In the isotropic case, we take $\mathbf{H} = \mathbf{I}$, otherwise we take $\mathbf{H} = \text{diag}(h_1, h_1, h_3)$ as previously described. The use of a smooth oscillatory model introduces the two parameters $\eta > 0$ and $\vartheta \in [0, 1)$ to be estimated.

### (iii) Negating of boundary effects

Neumann conditions on the boundary of $\mathcal{M}$ are known to introduce artefacts; essentially boundary regions are associated with higher variance [61]. This is undesirable, since there is no specific reason to expect the boundary of $\mathcal{M}$ to be more variable compared to the interior of $\mathcal{M}$. It has, however, been shown that the influence of boundary conditions vanishes exponentially quickly in the distance from the boundary [62]. To avoid such artefacts, in this paper, we work with panel data from the central region $[25, 275]^2$ of the domain $[0, 300]^2$ only and, in our experiments with the CHS, we extend the cylindrical manifold by 20 mm on either end.

### (c) Plane-stress elasto-plastic model

Consider a coupon which occupies the domain $\Omega \in \mathbb{R}^3$, with global coordinate system $(x, y, z) \in \mathbb{R}^3$, alongside a local coordinate system $(x_1, x_2, x_3) \in \mathbb{R}^3$. Here, the local coordinate $x_1$ is parallel to the layering at an angle $\theta$ to the global $x$ axis, $x_2$ is the build direction and $x_3 \equiv z$ is normal to the plane of the coupon.

For each experiment with a layer orientation $\theta$ to the horizontal axis $x$, the experiments provide longitudinal stress values $\sigma_L \equiv \sigma_y$ at given values of longitudinal strains $\varepsilon_L \equiv \varepsilon_y$. The connection from this experimental data to global material parameters/properties is a two-stage process.

Each experiment is modelled using a one-dimensional Ramberg–Osgood model,

$$\varepsilon_L = \frac{\sigma_L}{E_L(\theta)} + K\left(\frac{\sigma_L}{\sigma_0(\theta)}\right)^n. \tag{A 3}$$

This model is defined by three parameters $E_L(\theta)$, $\sigma_0(\theta)$ and $n$, where $K$ is taken to be the fixed constant 0.02. In modelling the overall material, since the $z$ dimension is relatively thin, plane-stress assumptions are imposed. In this section, the connection is set out between the independent Ramberg–Osgood parameters to 8 global material parameters, which are made up of 4 elastic material parameters $E_1$, $E_2$, $G_{12}$ and $\nu_{12}$, three independent stress factors $F$, $G$ and $N$ which define the onset of yield and a single ($\theta$ independent) hardening parameter, $n$.

The transversely anisotropic elastic properties of the material are defined uniquely by four parameters $E_1$, $E_2$, $G_{12}$ and $\nu_{12}$. The connection between local stresses and strains are defined by the compliance or stiffness matrices, given by

$$\mathbf{S} = \begin{bmatrix} \dfrac{1}{E_1} & -\dfrac{\nu_{12}}{E_2} & 0 \\[2mm] -\dfrac{\nu_{12}}{E_2} & \dfrac{1}{E_2} & 0 \\[2mm] 0 & 0 & \dfrac{1}{2G_{12}} \end{bmatrix} \quad \text{and} \quad \mathbf{C} = \mathbf{S}^{-1}, \tag{A 4}$$

respectively. Our controlled data from the experiments are measure values of stress in the longitudinal direction $\sigma_y \equiv \sigma_L$ at known longitudinal strain values $\varepsilon_y \equiv \varepsilon_L$. For the experimental set-up, the applied values of $\sigma_x$ and $\tau_{xy}$ are both zero. The resulting global strains, in global coordinates, can be calculated by rotating local coordinate properties by the layer orientation $\theta$, so that

$$\begin{bmatrix} \varepsilon_x \\ \varepsilon_y \\ \gamma_{xy} \end{bmatrix} = \mathbf{T}^{-1}\mathbf{S}\mathbf{T}\begin{bmatrix} 0 \\ \sigma_y \\ 0 \end{bmatrix} \quad \text{where } \mathbf{T} = \begin{bmatrix} c^2 & s^2 & 2sc \\ s^2 & c^2 & -2sc \\ -sc & sc & c \end{bmatrix}, \tag{A 5}$$

where $c = \cos(\theta)$ and $s = \sin(\theta)$. From this relationship, it follows that the elastic stiffness matrix in global coordinates is $\hat{\mathbf{C}} = \hat{\mathbf{S}}^{-1}$, whereby the longitudinal stiffness (the stiffness connecting longitudinal strains and stresses) is given by the entry $\hat{\mathbf{C}}_{22} = E_L$. This relationship provides a direct-mapping between material parameters $E_1$, $E_2$, $G_{12}$ and $\nu_{12}$ and the observed/controlled parameters $\theta$ and $E_L$. In general, the inverse of this map is ill-posed. However, in our setting data from experiments at multiple angles are available, enabling all parameters to in principle be identified.

The anisotropic yield response of the material is modelled with a classical Hill's potential function [53], which is defined by the yield surface as a functional of the local stress tensor $\sigma$ such that

$$f(\sigma) = \sqrt{F\sigma_{22}^2 + G\sigma_{11}^2 + H(\sigma_{11} - \sigma_{22})^2 + 2N\tau_{12}^2}, \quad \text{where } \sigma = \sigma_0(\theta)\mathbf{T}\begin{bmatrix} 0 \\ 1 \\ 0 \end{bmatrix}.$$

The onset of yield is the scalar value $\sigma_0(\theta)$ at which $f(\sigma) = 1$. Under plane stress assumptions the connection between the yield strength of $0°$, $90°$ and $45°$ coupons and the coefficients $F, G, H$ and $N$ are as follows:

$$F = \frac{1}{2}\frac{1}{\sigma_y(90°)^2}, \quad H = F, \quad G = \frac{1}{\sigma_y(0°)^2} - F \quad \text{and} \quad N = \frac{2}{\sigma_y(45°)^2} - \frac{1}{2}(F + G). \tag{A 6}$$

For the hardening exponent $n$ in the Ramberg–Osgood model, a single value across all experiments is inferred.

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
