## [Peer Review File · Proceedings. Mathematical, Physical, and Engineering Sciences]

Review History

RSPA-2021-0444.R0 (Original submission)

Review form: Referee 1

Is the manuscript an original and important contribution to its field?

Good

Is the paper of sufficient general interest?

Good

Is the overall quality of the paper suitable?

Good

Can the paper be shortened without overall detriment to the main message?

Yes

Do you think some of the material would be more appropriate as an electronic appendix?

No

Do you have any ethical concerns with this paper?

No

Recommendation?

Major revision is needed (please make suggestions in comments)

Comments to the Author(s)

This study is of interest to researchers in additive manufacturing. The anisotropic Gaussian process used in the study is robust, and has decent accuracy. The following comments are to be considered to improve the manuscript:

- 1) the title is too broad, and should be specified on mechanical property.
- 2) In Page 3 line 42, the authors claim "the geometry of WAAM steel depends on factors that are not easily measured or controlled". Please specify what factors.
- 3) The Design of Experiment should be described. The manufacturing parameters are not specified either. Otherwise, the quality of the training dataset is questionable.
- 4) Model Cross Validation: Test set performance was not included.
- 5) Why was the "natural gradient ascent" optimization metric chosen? Justification is required.
- 6) In table 1, parameters have not been normalised. The differing dimensionality might affect final results obtained?
- 7) In page 11 line 40-41, what sampling method was used?
- 8) Cross-validation was not discussed in the paper. The accuracy of the prediction should be specified by discussing the results of the model's application on the test data.

Review form: Referee 2

Is the manuscript an original and important contribution to its field?

Excellent

Is the paper of sufficient general interest?

Excellent

Is the overall quality of the paper suitable?

Excellent

Can the paper be shortened without overall detriment to the main message?

Yes

Do you think some of the material would be more appropriate as an electronic appendix?

No

Do you have any ethical concerns with this paper?

No

Recommendation?

Accept with minor revision (please list in comments)

Comments to the Author(s)

This is an excellent paper that, through taking into account the geometric variation that AM inevitably generates and accounting for anisotropy, should prove to be very helpful in driving forward understanding and quantification of uncertainty wrt. mechanical properties of AM produced components.

In my opinion one very minor revision is needed however - and this related to G12.

page 8 Line 24: E1 and E2 are entirely as would be expected for a steel as is Poisson ratio (154 and 148 GPa and 0.3 respectively) G12 however is far higher than would be expected - the Shear

modulus of steel being, typically, 60-80 GPa.

However it is done, the authors need to show that their model generates a physically meaningful value of G for steel. I'm certain that this will prove trivial to do - but it would be very helpful as the data here are sure to be cited and used.

Review form: Referee 3

Is the manuscript an original and important contribution to its field?

Excellent

Is the paper of sufficient general interest?

Excellent

Is the overall quality of the paper suitable?

Excellent

Can the paper be shortened without overall detriment to the main message?

Yes

Do you think some of the material would be more appropriate as an electronic appendix?

Yes

Do you have any ethical concerns with this paper?

No

Recommendation?

Accept with minor revision (please list in comments)

Comments to the Author(s)

This paper presents a generative statistical model that enables the quality of a design to be predicted before being manufactured. The paper is interesting and well written. It is of interest to 3D printing engineers. The paper can be accepted as it is. But some of the font sizes in the figures are too small to read. I suggest the readers increase them.

Decision letter (RSPA-2021-0444.R0)

20-Jul-2021

Dear Dr Dodwell

The Editor of Proceedings A has now received comments from referees on the above paper and would like you to revise it in accordance with their suggestions which can be found below (not including confidential reports to the Editor).

Please submit a copy of your revised paper within four weeks - if we do not hear from you within this time then it will be assumed that the paper has been withdrawn. In exceptional circumstances, extensions may be possible if agreed with the Editorial Office in advance.

Please note that it is the editorial policy of Proceedings A to offer authors one round of revision in which to address changes requested by referees. If the revisions are not considered satisfactory by

the Editor, then the paper will be rejected, and not considered further for publication by the journal. In the event that the author chooses not to address a referee's comments, and no scientific justification is included in their cover letter for this omission, it is at the discretion of the Editor whether to continue considering the manuscript.

To revise your manuscript, log into <http://mc.manuscriptcentral.com/prsa> and enter your Author Centre, where you will find your manuscript title listed under "Manuscripts with Decisions." Under "Actions," click on "Create a Revision." Your manuscript number has been appended to denote a revision.

You will be unable to make your revisions on the originally submitted version of the manuscript. Instead, revise your manuscript and upload a new version through your Author Centre.

When submitting your revised manuscript, you will be able to respond to the comments made by the referee(s) and upload a file "Response to Referees" in Step 1: "View and Respond to Decision Letter". Please provide a point-by-point response to the comments raised by the reviewers and the editor(s). A thorough response to these points will help us to assess your revision quickly. You can also upload a 'tracked changes' version either as part of the 'Response to reviews' or as a 'Main document'.

IMPORTANT: Your original files are available to you when you upload your revised manuscript. Please delete any unnecessary previous files before uploading your revised version.

When revising your paper please ensure that it remains under 28 pages long. In addition, any pages over 20 will be subject to a charge (£150 + VAT (where applicable) per page). Your paper has been ESTIMATED to be 21 pages.

Open Access

You are invited to opt for open access, our author pays publishing model. Payment of open access fees will enable your article to be made freely available via the Royal Society website as soon as it is ready for publication. For more information about open access please visit <https://royalsociety.org/journals/authors/open-access/>. The open access fee for this journal is £1700/\$2380/€2040 per article. VAT will be charged where applicable. Please note that if the corresponding author is at an institution that is part of a Read and Publishing deal you are required to select this option. See <https://royalsociety.org/journals/librarians/purchasing/read-and-publish/read-publish-agreements/> for further details.

Once again, thank you for submitting your manuscript to Proc. R. Soc. A and I look forward to receiving your revision. If you have any questions at all, please do not hesitate to get in touch.

Yours sincerely
Raminder Shergill
proceedingsa@royalsociety.org

on behalf of
Professor Yihui Zhang
Board Member
Proceedings A

Reviewer(s)' Comments to Author:

Referee: 1

Comments to the Author(s)

This study is of interest to researchers in additive manufacturing. The anisotropic Gaussian process used in the study is robust, and has decent accuracy. The following comments are to be considered to improve the manuscript:

- 1) the title is too broad, and should be specified on mechanical property.
- 2) In Page 3 line 42, the authors claim "the geometry of WAAM steel depends on factors that are not easily measured or controlled". Please specify what factors.
- 3) The Design of Experiment should be described. The manufacturing parameters are not specified either. Otherwise, the quality of the training dataset is questionable.
- 4) Model Cross Validation: Test set performance was not included.
- 5) Why was the "natural gradient ascent" optimization metric chosen? Justification is required.
- 6) In table 1, parameters have not been normalised. The differing dimensionality might affect final results obtained?
- 7) In page 11 line 40-41, what sampling method was used?
- 8) Cross-validation was not discussed in the paper. The accuracy of the prediction should be specified by discussing the results of the model's application on the test data.

Referee: 2

Comments to the Author(s)

This is an excellent paper that, through taking into account the geometric variation that AM inevitably generates and accounting for anisotropy, should prove to be very helpful in driving forward understanding and quantification of uncertainty wrt. mechanical properties of AM produced components.

In my opinion one very minor revision is needed however - and this related to G12.

page 8 Line 24: E1 and E2 are entirely as would be expected for a steel as is Poisson ratio (154 and 148 GPa and 0.3 respectively) G12 however is far higher than would be expected - the Shear modulus of steel being, typically, 60-80 GPa.

However it is done, the authors need to show that their model generates a physically meaningful value of G for steel. I'm certain that this will prove trivial to do - but it would be very helpful as the data here are sure to be cited and used.

Referee: 3

Comments to the Author(s)

This paper presents a generative statistical model that enables the quality of a design to be predicted before being manufactured. The paper is interesting and well written. It is of interest to 3D printing engineers. The paper can be accepted as it is. But some of the font sizes in the figures are too small to read. I suggest the readers increase them.

Author's Response to Decision Letter for (RSPA-2021-0444.R0)

See Appendix A.

RSPA-2021-0444.R1 (Revision)

Review form: Referee 1

Is the manuscript an original and important contribution to its field?

Acceptable

Is the paper of sufficient general interest?

Acceptable

Is the overall quality of the paper suitable?

Acceptable

Can the paper be shortened without overall detriment to the main message?

Yes

Do you think some of the material would be more appropriate as an electronic appendix?

No

Do you have any ethical concerns with this paper?

No

Recommendation?

Accept as is

Comments to the Author(s)

No other comments

Review form: Referee 2

Is the manuscript an original and important contribution to its field?

Excellent

Is the paper of sufficient general interest?

Excellent

Is the overall quality of the paper suitable?

Excellent

Can the paper be shortened without overall detriment to the main message?

Yes

Do you think some of the material would be more appropriate as an electronic appendix?

No

Do you have any ethical concerns with this paper?

No

Recommendation?

Accept as is

Comments to the Author(s)

I am happy with the changes made - and somewhat relieved i didn't send you on a wild goose chase. Those figures all look reasonable now.

Review form: Referee 3

Is the manuscript an original and important contribution to its field?

Good

Is the paper of sufficient general interest?

Good

Is the overall quality of the paper suitable?

Excellent

Can the paper be shortened without overall detriment to the main message?

Yes

Do you think some of the material would be more appropriate as an electronic appendix?

No

Do you have any ethical concerns with this paper?

No

Recommendation?

Accept as is

Comments to the Author(s)

The authors have addressed all the comments.

Decision letter (RSPA-2021-0444.R1)

08-Oct-2021

Dear Dr Dodwell

I am pleased to inform you that your manuscript entitled "A Data-Centric Approach to Generative Modelling for 3D-Printed Steel" has been accepted in its final form for publication in Proceedings A.

Our Production Office will be in contact with you in due course. You can expect to receive a proof of your article soon. Please contact the office to let us know if you are likely to be away from e-mail in the near future. If you do not notify us and comments are not received within 5 days of sending the proof, we may publish the paper as it stands.

As a reminder, you have provided the following 'Data accessibility statement' (if applicable). Please remember to make any data sets live prior to publication, and update any links as needed when you receive a proof to check. It is good practice to also add data sets to your reference list.
Statement (if applicable):

Open access

You are invited to opt for open access, our author pays publishing model. Payment of open access fees will enable your article to be made freely available via the Royal Society website as soon as it is ready for publication. For more information about open access please visit <https://royalsociety.org/journals/authors/which-journal/open-access/>. The open access fee for this journal is £1700/\$2380/€2040 per article. VAT will be charged where applicable.

Note that if you have opted for open access then payment will be required before the article is published – payment instructions will follow shortly.

If you wish to opt for open access then please inform the editorial office (proceedingsa@royalsociety.org) as soon as possible.

Your article has been estimated as being 22 pages long. Our Production Office will inform you of the exact length at the proof stage.

Proceedings A levies charges for articles which exceed 20 printed pages. (based upon approximately 540 words or 2 figures per page). Articles exceeding this limit will incur page charges of £150 per page or part page, plus VAT (where applicable).

Under the terms of our licence to publish you may post the author generated postprint (ie. your accepted version not the final typeset version) of your manuscript at any time and this can be made freely available. Postprints can be deposited on a personal or institutional website, or a recognised server/repository. Please note however, that the reporting of postprints is subject to a media embargo, and that the status the manuscript should be made clear. Upon publication of the definitive version on the publisher's site, full details and a link should be added.

You can cite the article in advance of publication using its DOI. The DOI will take the form: 10.1098/rspa.XXXX.YYYY, where XXXX and YYYY are the last 8 digits of your manuscript number (eg. if your manuscript number is RSPA-2017-1234 the DOI would be 10.1098/rspa.2017.1234).

For tips on promoting your accepted paper see our blog post:
<https://royalsociety.org/blog/2020/07/promoting-your-latest-paper-and-tracking-your-results/>

On behalf of the Editor of Proceedings A, we look forward to your continued contributions to the Journal.

Sincerely,
Raminder Shergill
proceedingsa@royalsociety.org

on behalf of
Professor Yihui Zhang
Board Member
Proceedings A

Reviewer(s)' Comments to Author:

Referee: 2

Comments to the Author(s)

I am happy with the changes made - and somewhat relieved i didn't send you on a wild goose chase. Those figures all look reasonable now.

Referee: 1

Comments to the Author(s)

no other comments

Referee: 3

Comments to the Author(s)

The authors have addressed all the comments.

Appendix A

Reply to Reviewers - A Data-Centric Approach to Generative Modelling for 3D-Printed Steel

We would like to thank the referees for their positive comments of the manuscript and their helpful suggestions and questions to improve the clarity of our work.

Reviewer 1

Q1. *The title is too broad, and should be specified on mechanical property.*

Reply to Reviewers. We believe the paper presents a general study of generative modelling of 3D printed steel. Whilst we do not include all aspects of 3D-Printed steel, we include more than just the mechanical properties. For example uncertain manufacturing defects (geometric), material properties and the resulting structural behaviour. We therefore request to keep the title more general as ‘A Data-Centric Approach to Generative Modelling of 3D-Printed Steel’.

Q2 *On Page 3 line 42, the authors claim “the geometry of WAAM steel depends on factors that are not easily measured or controlled”. Please specify what factors.*

Reply to Reviewers. Thank you for this comment; we have improved the text to make these factors clearer. The original paragraph

‘The geometry of WAAM steel depends on factors that are not easily measured or controlled, motivating the treatment of material geometry as a random variable whose statistical properties can in principle be described.’

now reads as follows:

‘The final geometry of WAAM steel is treated as a random variable, whose statistical properties can in principle be described, due to the complex interaction between the process input parameters, including the welding voltage and current, robot arm speed and position, layer temperature, ability to cool between welding passes and part thickness.’

Q3. *The Design of Experiment should be described. The manufacturing parameters are not specified either. Otherwise, the quality of the training dataset is questionable.*

Reply to Reviewers. The experimental protocol has now been included, in the new Appendix (a), and referenced from the main text:

‘For building the components that comprise the training dataset, wire of 1.0 mm diameter was used while the employed welding speed and wire feed rate were 15-30 mm/s and 4-8 m/min respectively. The employed shield gas was 98% AR and 2% CO₂ at a flow rate of 10-20 L/min, the current and arc voltage of the deposition process were 100-140 A and 18-21 V respectively, while the deposition rate was between 0.5-2.0 kg/h.’

Q4. & Q8. *Here we address two comments from the same reviewer together, these are (4) Model Cross Validation: Test set performance was not included. (8) Cross-validation was not discussed in the paper. The accuracy of the prediction should be specified by discussing the results of the model’s application on the test data.*

Reply to Reviewers. There appears to be some confusion on this point, for which we apologise and will seek to avoid in the manuscript: In settings where one is attempting to predict a dependent variable (y) given one or more independent variables (x), cross-validation can be used to *test* how well y is predicted using knowledge of x and a statistical model. This is not our setting; there is no analogue of x in our generative statistical model (except perhaps, in an abstract sense, where x represents the notional geometry of the component – but then we have only two different instances of x in our dataset). Our statistical generative model is validated in Section 4(b), where predictions of the mechanical performance of a circular hollow section (CHS) are tested in detail. It may be that confusion has arisen because cross-validation is also widely used to *train* statistical models, and our generative statistical model also needs to be trained. However, several equally valid approaches to training exist, and in this paper we employed maximum (marginal) likelihood. Maximum (marginal) likelihood was used for this work because it facilitates straight-forward model selection via the Akaike information criterion (AIC). Note that the (marginal) likelihood is formally equivalent to training by exhaustive leave-p-out cross-validation averaged over all values of p and all held-out test sets when using the log posterior predictive probability as the scoring rule; see Fong E, Holmes CC. On the marginal likelihood and cross-validation. *Biometrika*. 2020 Jun 1;107(2):489-96. The text in Section 2(b) of the manuscript has been clarified as follows:

‘To assess which combination of these geometric features best describes WAAM steel, we first fitted each model to the training dataset and then assessed goodness-of-fit in light of the complexity of each model. For training, maximum likelihood was employed due to its relatively low computational cost (e.g. compared to cross validation) and because it facilitates model selection under an Occam’s razor principle, via the Akaike information criterion [...]’

Q5. *Why was the ‘natural gradient ascent’ optimization metric chosen? Justification is required.*

Reply to Reviewers. Apologies for the confusion on this point; the objective function that is maximised is the marginal likelihood (which is standard), and “natural gradient ascent” is a numerical algorithm that can be employed to seek a maximiser of the marginal likelihood. This particular numerical algorithm was chosen because it can usefully exploit the conditional Gaussian structure present in our statistical model in order to quickly locate a maximiser of the marginal likelihood. To avoid this confusion, in Section 2(b) of the revised manuscript we now write:

‘To numerically approximate the maximiser $\hat{\phi}$ of the likelihood in (2.4) we exploit an iterative numerical optimisation method called ‘natural gradient ascent’ [...]’

Q6. *In table 1, parameters have not been normalised. The differing dimensionality might affect final results obtained?*

Reply to Reviewers. The Markov Chain Monte Carlo (MCMC) computations used an adaptive Metropolis proposal distribution, where the initial covariance of the proposal was scaled proportional to the variance of the priors for each quantities. This has the affect of rescaling space, and preventing any issue in terms of convergence of the MCMC strategy. Diagnostic calculations for all MCMC runs, which include cross-validation against multiple chains, were used to ensure convergence to the true posterior distributions. This has been described in Section 3(a).

Q7. *In page 11 line 40-41, what sampling method was used?*

Reply to Reviewers. Random samples were drawn independently and uniformly from the MCMC output. A note has been added to the paper to make this clear. The line now reads:

‘Sampled material parameters, randomly drawn from the set of posterior samples generated in section 3, can be input into this FE material model.’

Reviewer 2

Q1. *E_1 and E_2 are entirely as would be expected for a steel as is Poisson ratio (154 and 148 GPa and 0.3 respectively) G_{12} however is far higher than would be expected - the Shear modulus of steel being, typically, 60-80 GPa. However it is done, the authors need to show that their model generates a physically meaningful value of G for steel. I’m certain that this will prove trivial to do - but it would be very helpful as the data here are sure to be cited and used.*

Reply to Reviewers. We thank the reviewer for spotting this presentation error in our results. We note that we have actually estimate $2G_{12}$, because of engineering voigt notation this actually halves the ”modulus”. We have corrected this point throughout the paper, and notably in the appendix section (b) where we derive the specific calculations. This has no impact on the results presented in the paper. This gives the a mean value as $G_{12} = 108\text{GPa}$, for which we should get close to the corresponding shear value for a single crystal, which for Fe is about 120GPa, hence this seems reasonable.

Reviewer 3

Questions from Reviewer *No technical questions raised by reviewer. The reviewer asked the authors to review the size of font in the some of the figures.*

Reply to Reviewers. *Thank you for this feedback. Text in Fig. 4 and Fig. 6; have been made larger to improve readability. In particular Fig. 6 has been split in two separate figures to improve text size and presentation.*